# Biological Screening and Crystallographic Studies of Hydroxy γ-Lactone Derivatives to Investigate PPARγ Phosphorylation Inhibition

**DOI:** 10.3390/biom13040694

**Published:** 2023-04-19

**Authors:** Davide Capelli, Giulia Cazzaniga, Matteo Mori, Antonio Laghezza, Fulvio Loiodice, Martina Quaglia, Elisa Negro, Fiorella Meneghetti, Stefania Villa, Roberta Montanari

**Affiliations:** 1Istituto di Cristallografia, Consiglio Nazionale delle Ricerche, Strada Provinciale 35d, n. 9-00010, Montelibretti, 34149 Rome, Italy; 2Department of Pharmaceutical Sciences, University of Milan, Via L. Mangiagalli 25, 20133 Milano, Italy; 3Department of Pharmacy—Drug Sciences, University of Bari “Aldo Moro”, Via Orabona 4, 70125 Bari, Italy

**Keywords:** X-ray crystallography, drug design, heterocycle, PPARγ phosphorylation

## Abstract

PPARγ represents a key target for the treatment of type 2 diabetes and metabolic syndrome. To avoid serious adverse effects related to the PPARγ agonism profile of traditional antidiabetic drugs, a new opportunity is represented by the development of molecules acting as inhibitors of PPARγ phosphorylation by the cyclin-dependent kinase 5 (CDK5). Their mechanism of action is mediated by the stabilization of the PPARγ β-sheet containing Ser273 (Ser245 in PPARγ isoform 1 nomenclature). In this paper, we report the identification of new γ-hydroxy-lactone-based PPARγ binders from the screening of an in-house library. These compounds exhibit a non-agonist profile towards PPARγ, and one of them prevents Ser245 PPARγ phosphorylation by acting mainly on PPARγ stabilization and exerting a weak CDK5 inhibitory effect.

## 1. Introduction

The role of peroxisome proliferator-activated receptors (PPARs) in several chronic diseases, such as type 2 diabetes (T2D), obesity, and atherosclerosis, is well-known [1,2,3]. Among the three subtypes (α, β/δ, and γ), PPARγ is the most widely studied as a therapeutically attractive target because of its key role in the regulation of energy balance and fat cell differentiation in adipose tissue [3,4,5,6,7,8]. PPARγ is a ligand-activated transcription factor which forms a heterodimer with the retinoid X receptor (RXR) and binds to specific promoter response elements (PPREs) on the DNA, acting as a gene expression regulator [9].

PPARγ takes part in the control of many cellular functions and pathways related to the regulation of fatty acid metabolism and glucose homeostasis, in particular, adipokine gene expression control and promotion of adiponectin biosynthesis. Notably, serum adiponectin levels are decreased in patients with obesity, insulin resistance, cardiovascular diseases, and hypertension compared to healthy subjects [10].

Unfortunately, only a small number of PPARγ ligands are in Phase II trials. Thiazolidinediones (TZDs), including rosiglitazone and pioglitazone, are the only clinically available agents controlling hyperglycemia by improving insulin resistance, which is regarded as a hallmark and early etiologic basis for T2D. TZDs have been demonstrated to behave as PPARγ full agonists, activating PPARγ and leading to a wide spectrum of gene expression changes, resulting in insulin-sensitizing effects [11]. However, further studies demonstrated that the use of these compounds is associated with unwanted effects, such as weight gain, fluid retention, increased incidence of cardiovascular events, and bone fractures [12,13,14,15]. Hence, over the years, various strategies have been attempted to obtain compounds with good therapeutic potential and fewer side effects. Among the possible candidates are PPARα-δ/γ dual agonists [16,17] or PPARα/γ/δ pan-agonists [18], which beneficially alter carbohydrate and lipid metabolism in a coordinated manner, and selective PPARγ modulators (SPPARγMs). The latter showed an improved therapeutic profile compared to PPARγ full agonists, due to the different conformational changes induced by the ligands, giving rise to different PPARγ transcriptional signatures [19,20,21,22,23,24]. In this regard, molecules with an antagonist profile towards PPARγ should not be overlooked [25,26,27,28,29] (bexarotene, 2-phenylamino pyrimidine, and *N*-biphenylmethylindole derivatives). These antagonists maintain robust antidiabetic activity in rodent models of diabetes and may provide a safe alternative to targeting PPARγ for the therapeutic intervention in insulin resistance and T2D.

Another strategy for designing new PPARγ ligands exploits the great structural plasticity of this nuclear receptor, whose functions are modulated by the fluctuations of its three-dimensional structure. The latter is mainly regulated by ligand binding, which induces structural changes, thus affecting the interactions with co-activators or co-repressors to stimulate or inhibit their functions. PPARγ activities are also regulated by various post-translational modifications, including phosphorylation, SUMOylation, ubiquitination, acetylation, and O-GlcNAcylation, which are found at numerous modification sites and have been associated with the status of metabolic diseases [30].

The dissection of the interplay between these post-translational modifications could then provide a new strategy for designing novel PPARγ ligands able to modulate PPARγ functions through the fine modulation of protein folding.

In 2010, Choi and co-workers [19,20] described the clear correlation between high-fat diets in vivo and the cyclin-dependent kinase 5 (CDK5)-mediated phosphorylation of PPARγ at Ser273 (Ser245 in PPARγ isoform 1 nomenclature). On this basis, they gave rise to a new class of non-agonist antidiabetic compounds, promoting the loss of insulin sensitivity but devoid of the typical side effects associated with PPARγ agonists. In 2014 the same authors demonstrated that the phosphorylation of PPARγ-Ser245 by CDK5 did not alter its adipogenic activity but dysregulated a specific set of genes with roles in obesity and diabetes. A clear example is represented by the non-agonist PPARγ ligand SR1664, which showed an impressive potential as an inhibitor of PPARγ phosphorylation while exerting potent anti-diabetic activity. Unlike TZDs and other PPARγ agonists, this compound did not cause fluid retention or weight gain in vivo or reduce osteoblast mineralization in culture.

It is well known that CDK5 activity increases in inflammatory conditions, such as obesity (Figure 1A). Ser245 phosphorylation by CDK5 results in the dysregulation of a subset of PPARγ target genes that are known to be associated with tissue insulin sensitization, whereas its inhibition maintains the transcription of several of these genes (and the correlated insulin sensitization) unchanged.

The CDK5-mediated phosphorylation of PPARγ does not have any influence either on its transactivation activity or on its role in adipocyte differentiation, meaning that any ligand acting as an inhibitor of this phosphorylation mechanism would be able to exert antidiabetic effects. In our previous studies, we reported the inhibition mechanism of PPARγ phosphorylation by CDK5 [31]. In detail, we elucidated the structural mechanism through which the kinase, in complex with its coactivator p25, is able to gain access to the PPARγ phosphorylation site, and we described what happens at a structural level during PPARγ phosphorylation (Figure 1B). Further studies also showed that partial distortion of the β-4 strand makes PPARγ more susceptible to phosphorylation and that PPARγ ligands serve to stabilize this β-strand, thus making this region of the ligand binding domain (LBD) less available for energetically plausible binding with a kinase, regardless of the degree of ligand agonism or antagonism [29]. This new mechanistic scenario might explain why some PPARγ partial agonists or modulators can exhibit antidiabetic effects similar to TZDs and other full agonists, without showing their typical side effects.

In this work, we will discuss the identification of new inhibitors of the CDK5-mediated phosphorylation of PPARγ from the biological screening of an in-house library of γ-hydroxy-lactone derivatives. This scaffold constitutes the cyclic core of butyrolactone I ((*R*)-methyl 2-(4-hydroxy-2-(4-hydroxy-3-(3-methylbut-2-en-1-yl)benzyl)-3-(4-hydroxyphenyl)-5-oxo-2,5-dihydrofuran-2-yl)acetate) (BLI, Figure 2), a natural product, isolated from *Aspergillus terreus*, that was reported in the literature as a PPARγ partial agonist [10].

The γ-hydroxy-lactone nucleus has been reported in the literature as a core structural element of several bioactive compounds, such as anti-inflammatory, antioxidant, and antitubercular agents [32,33,34]. Interestingly, BLI showed dual modulator activities as both a CDK5 inhibitor and a PPARγ partial agonist. Based on the structural similarity of our compound collection to BLI, we decided to search for innovative PPARγ binders bearing a pentacyclic lactone core. We started by evaluating the affinities of an initial pool of 30 molecules sharing this scaffold. The affinity of the potential ligands to PPARγ was evaluated by surface plasmon resonance (see Appendix A); among the tested molecules, **1**, **2**, and **3** (Figure 2) emerged as the most interesting for further investigations. Therefore, these compounds were analyzed for their in vitro activity (agonist and/or antagonist assay) towards PPARγ receptors by using the GAL4-PPAR transactivation assay. Moreover, they were tested in vitro for the inhibition of CDK5-mediated PPARγ phosphorylation. In addition, X-ray studies were performed on the complexes between PPARγ and the ligands to reveal the specific binding mode of this class of compounds. Through our efforts, we confirmed that the γ-lactone skeleton can be considered as a promising scaffold to design novel PPARγ binders.

## 2. Materials and Methods

### 2.1. Chemistry

All starting reagents and solvents were purchased from commercial suppliers (Merck KGaA, Darmstadt, Germany; FluoroChem, Hadfield, UK) and used as received. Anhydrous solvents were utilized without further drying. The course of the reactions was followed by thin-layer chromatography (TLC) using aluminum-backed silica gel 60 plates (0.2 mm; Merck KGaA). Crude products were purified by flash column chromatography on silica gel 60 (40–63 µM; Merck KGaA), using the indicated solvent system. Melting points were determined in open capillary tubes with a Stuart SMP30 Melting Point Apparatus (Cole-Parmer Stuart, Stone, UK). All tested compounds were characterized by means of one-dimensional NMR techniques and high-resolution mass spectrometry (HRMS). ^1^H and ^13^C NMR spectra were acquired at ambient temperature with a Varian Oxford 300 MHz instrument (Varian, Palo Alto, CA, USA) operating at 300 MHz for ^1^H and 75 MHz for ^13^C. Chemical shifts are expressed in ppm (δ), and *J*-couplings are given in Hertz. HRMS analyses were carried out on a Q-ToF Synapt G2-Si HDMS system (Waters, Milford, MA, USA). All relevant spectra are reported in the Appendix A.

All the final compounds were synthesized as a racemic mixture. Compound **1** (Figure 1) was obtained starting from the acetylation of the commercially available 2-aminoacetic acid using acetic anhydride [35]. The resulting intermediate (**4**) was cyclized with 4-bromobenzaldehyde in acetic anhydride and sodium acetate to afford the oxazolone derivative **5** [36]. A hydrolysis reaction in 3 M HCl led to intermediate **6**, which was subjected to a Fischer–Speier esterification using ethanol and concentrated HCl to give compound **7**. The final product **1** was obtained by a cyclization reaction between the phenylpyruvic ester derivative **7** and 3-hydroxybenzaldehyde.

Compound **2** (Figure 2) was synthesized by a one-pot reaction between acetophenone and diethyl oxalate to obtain intermediate **8**, which was then subjected to a cyclization reaction with 3-hydroxybenzaldehyde to afford the desired product **2**.

Compound **3** (Figure 3) was obtained by the same cyclization reaction used for **1**, starting from methyl phenylpyruvate (**9**) and 4-(naphthalen-1-ylmethoxy)benzaldehyde (**10**). The former was synthesized by the esterification of phenylpyruvic acid using methyl chloride and DBU [37]. The latter was afforded by a substitution reaction between 4-hydroxybenzaldehyde and 1-(chloromethyl)naphthalene, in the presence of oven-dried K_2_CO_3_ [38].

2-Acetamidoacetic acid (**4**). Acetic anhydride (7.18 mmol, 2.7 eq.) was added to a solution of 2-aminoacetic acid (2.66 mmol, 1 eq.) in MeOH (1 mL), and the mixture was refluxed for 6 h. Then, the volatiles were evaporated in vacuo, and the residue was washed with MeOH to afford the desired product (**4**) as a white solid. Yield: 75%. Mp: 206–207 °C. TLC (dichloromethane/methanol 8:2)—R_f_: 0.60. ^1^H NMR (300 MHz, CD_3_OD) δ (ppm): δ 3.89 (s, 2 H, CH_2_), 1.99 (s, 3 H, CH_3_).

(Z)-4-(4-Bromobenzylidene)-2-methyloxazol-5(4H)-one (**5**). A solution of 2-acetamidoacetic acid (**4**, 1.71 mmol, 1.1 eq.), 4-bromobenzaldehyde (1.55 mmol, 1 eq.), and NaOAc (1.55 mmol, 1 eq.) in acetic anhydride (0.5 mL) was stirred at 100 °C for 5 h. The reaction mixture was cooled to room temperature, and the residue was filtered and washed with MeOH to give a yellow solid. Yield: 95%. Mp: 196–198 °C (dec.). TLC (cyclohexane/EtOAc 9:1)—R_f_: 0.53. ^1^H NMR (300 MHz, CDCl_3_) δ (ppm) 7.95 (d, *J* = 8.4 Hz, 2 H, H_Ar_), 7.57 (d, *J* = 8.6 Hz, 2 H, H_Ar_), 7.06 (s, 1 H, CH), 2.41 (s, 3 H, CH_3_).

2-Acetamido-3-(4-bromophenyl)propanoic acid (**6**). A solution of (*Z*)-4-(4-bromobenzylidene)-2-methyloxazol-5(4*H*)-one (**5**, 0.60 mmol) in 3M HCl (0.6 mL) was stirred at reflux for 3 h. The solid was filtered off to obtain compound **6** as a yellow solid. Yield: 80%. Mp: 216–218 °C (dec.). TLC (DCM/MeOH 8:2)—R_f_: 0.13. ^1^H NMR (300 MHz, CD_3_OD) δ (ppm) 7.55 (d, *J* = 8.6 Hz, 2 H, H_Ar_), 7.47 (d, *J* = 8.6 Hz, 2 H, H_Ar_), 7.41 (s, 1 H, CH), 2.09 (s, 3 H, CH_3_).

Methyl 3-(4-bromophenyl)-2-oxopropanoate (**7**). Concentrated HCl (1.40 mL) was added dropwise to a solution of 2-acetamido-3-(4-bromophenyl)propanoic acid (**6**, 0.71 mmol) in EtOH (1.4 mL), and the reaction mixture was stirred at reflux for 4 h. The solvent was then removed in vacuo at room temperature, and the resulting solution was extracted with EtOAc (3 × 3 mL). The organic phase was dried over anhydrous Na_2_SO_4_, filtered, and evaporated under reduced pressure. The desired intermediate (**7**) was not isolated, and the crude brown liquid was directly used for the next synthetic step. TLC (cyclohexane/EtOAc 6:4)—R_f_: 0.61.

4-(4-Bromophenyl)-3-hydroxy-5-(3-hydroxyphenyl)furan-2(5H)-one (**1**). DBU (0.75 mmol, 1.01 eq.) was added dropwise to a cold solution of methyl 3-(4-bromophenyl)-2-oxopropanoate (**7**, 0.74 mmol, 1 eq.) and 3-hydroxybenzaldehyde (0.96 mmol, 1.3 eq.) in dry DMF (3.1 mL), and the reaction mixture was stirred at 0 °C for 5 h under N_2_ atmosphere. The resulting solution was then diluted with EtOAc (10 mL) and extracted with H_2_O (3 × 15 mL). The aqueous phase was acidified with 3 M HCl and extracted with EtOAc (3 × 20 mL). The organic phase was washed with cold H_2_O to remove the residual DMF, dried over anhydrous Na_2_SO_4_, filtered, and evaporated in vacuo. The resulting solid was purified by flash column chromatography (cyclohexane/ EtOAc 6:4) to afford the final compound (**1**) as a white solid. Yield: 50%. Mp: 194–195 °C (dec.). TLC (cyclohexane/EtOAc 6:4)—R_f_: 0.18. ^1^H NMR (300 MHz, CD_3_OD) δ (ppm) 7.59–7.49 (m, 2 H, H_Ar_), 7.49–7.39 (m, 2 H, H_Ar_), 7.17 (t, *J* = 7.8 Hz, 1 H, H_Ar_), 6.89–6.69 (m, 3 H, H_Ar_), 6.21 (s, 1 H, CH). ^13^C NMR (75 MHz, CD_3_OD) δ (ppm) 169.64, 157.76, 139.43, 137.54, 131.17, 129.78, 129.74, 129.71, 128.97, 126.66, 121.93, 118.96, 116.20, 114.11, 80.66. HRMS (ESI/Q-ToF): calcd. for C_16_H_11_BrO_4_ 344.9762, found 344.9762 [M-H]^−^.

4-Benzoyl-3-hydroxy-5-(3-hydroxyphenyl)furan-2(5H)-one (**2**). Initially, Na was slowly added to dry MeOH under a N_2_ atmosphere to obtain a 2 M solution of NaOMe. Acetophenone (1.6 mmol, 1 eq.) and the freshly prepared aliquot of 2 M NaOMe (1 mL) were stirred in dry MeOH (1.4 mL) at room temperature for 2 h under a N_2_ atmosphere. Then, diethyl oxalate (1.75 mmol, 1.1 eq.) was added dropwise to the solution, and the reaction mixture was stirred at reflux for 3 h. Intermediate **8** was not isolated, and the obtained solution was cooled to room temperature and diluted with H_2_O (1 mL). 3-Hydroxybenzaldehyde (1.6 mmol, 1 eq.) was added to the reaction, and the mixture was stirred at room temperature overnight. The solution was diluted with EtOAc (3 mL) and extracted with H_2_O (3 × 5 mL). The aqueous phase was acidified with 3 M HCl and extracted with EtOAc. The organic phase was dried over anhydrous Na_2_SO_4_, filtered, and evaporated in vacuo. The resulting solid was purified by recrystallization using a 1:1 mixture of EtOH/H_2_O. The final compound (**2**) was obtained as a white solid. Yield: 20%. Mp: 181.5–182.5 °C (dec.). TLC (DCM/MeOH)—R_f_: 0.43. ^1^H NMR (300 MHz, CD_3_OD) δ (ppm) 7.86–7.77 (m, 2 H, H_Ar_), 7.63–7.51 (m, 1 H, H_Ar_), 7.51–7.39 (m, 2 H, H_Ar_), 7.14 (t, *J* = 7.8 Hz, 1 H, H_Ar_), 6.88–6.74 (m, 2 H, H_Ar_), 6.74–6.68 (m, 1 H, H_Ar_), 6.27 (s, 1 H, CH). ^13^C NMR (75 MHz, CD_3_OD) δ (ppm) 189.86, 169.09, 157.47, 144.77, 137.05, 136.86, 133.09, 129.43, 128.82, 128.04, 125.16, 118.05, 115.88, 113.57, 80.82. HRMS (ESI/Q-ToF): calcd. for C_17_H_12_O_5_ 295.0606, found 295.0606 [M-H]^−^.

(Z)-Methyl 2-hydroxy-3-phenylacrylate (**9**). 1,8-Diazobicyclo(5.4.0)undec-7-ene (DBU) (0.61 mmol, 1 eq.) and iodomethane (3.05 mmol, 5 eq.) were added to a solution of pyruvic acid (0.61 mmol, 1 eq.) in dry DMF (3 mL) at 0 °C under a N_2_ atmosphere, and the resulting mixture was stirred at the same temperature for 2.5 h. The reaction solution was acidified with 1M HCl (3 mL) and extracted with Et_2_O (3 × 10 mL). The organic phase was dried over anhydrous Na_2_SO_4_, filtered, and evaporated in vacuo. The resulting solid was purified by recrystallization using a 1:1 mixture of EtOH/H_2_O to afford compound **9** as a white solid. Yield: 55%. Mp: 58 °C. TLC (cyclohexane/EtOAc 8:2)—R_f_: 0.47. ^1^H-NMR (300 MHz, CDCl_3_) δ (ppm): δ 7.78–7.75 (m, 2 H, H_Ar_), 7.40–7.28 (m, 3 H, H_Ar_), 6.53 (s, 1 H, CH), 6.40 (bs exch. D_2_O, 1 H, OH), 3.92 (s, 3 H, CH_3_).

4-(Naphthalen-1-ylmethoxy)benzaldehyde (**10**). 1-(Chloromethyl)naphthalene (1.96 mmol, 1.2 eq.) was added to a stirred suspension of 4-hydroxybenzaldehyde (0.98 mmol, 1 eq.) and oven-dried K_2_CO_3_ (3.28 mmol, 4 eq.) in dry CH_3_CN (3 mL) at room temperature. The mixture was stirred at reflux (~80 °C) for 4 h, under a N_2_ atmosphere. Then, the reaction was extracted with EtOAc (3 × 5 mL), and the organic phase was dried over anhydrous Na_2_SO_4_, filtered, and evaporated in vacuo. The resulting solid was purified by flash column chromatography (cyclohexane/EtOAc 9:1) to afford the desired compound (**10**) as a white solid. Yield: 60%. Mp: 101 °C. TLC (cyclohexane/EtOAc 9:1)—R_f_: 0.24. ^1^H NMR (300 MHz, CDCl_3_) δ (ppm): δ 9.91 (s, 1 H, H_Ar_), 8.04–8.01 (m, 1 H, H_Ar_), 7.94–7.86 (m, 4 H, H_Ar_), 7.61–7.46 (m, 4 H, H_Ar_), 7.16 (d, *J* = 8.7 Hz, 2 H, H_Ar_), 5.59 (s, 2 H, CH_2_).

3-Hydroxy-5-(4-(naphthalen-1-ylmethoxy)phenyl)-4-phenylfuran-2(5H)-one (**3**). (Z)-Methyl 2-hydroxy-3-phenylacrylate (**9**, 1.10 mmol, 1 eq.) and DBU (1.10 mmol, 1.01 eq.) were added dropwise to a cold solution of 4-(naphthalen-1-ylmethoxy)benzaldehyde (**10**, 3.3 mmol, 1.3 eq.) in dry DMF (5 mL), and the reaction was stirred at 0 °C for 2 h, under N_2_ atmosphere. The mixture was acidified with 3M HCl and extracted with EtOAc (3 × 10 mL). The organic phase was dried over anhydrous Na_2_SO_4_, filtered, and evaporated in vacuo. The resulting solid was purified by flash column chromatography (cyclohexane/EtOAc 7:3) to afford the final compound (**3**) as a white solid. Yield: 30%. Mp: 204 °C. TLC (cyclohexane/EtOAc 8:2)—R_f_: 0.10. ^1^H NMR (300 MHz, CDCl_3_) δ (ppm): δ 8.02–7.99 (m, 1 H, H_Ar_), 7.91–7.85 (m, 2 H, H_Ar_), 7.61–7.43 (m, 6 H, H_Ar_), 7.37–7.29 (m, 5 H, H_Ar_), 7.03 (d, *J* = 8.7 Hz, 2 H, H_Ar_), 6.28 (bs exch. D_2_O, 1 H, OH), 6.22 (s, 1 H, CH), 5.47 (s, 2 H, CH_2_). ^13^C NMR (75 MHz, CDCl_3_) δ (ppm): 170.00, 159.94, 137.56, 133.78, 131.82, 131.47, 129.90, 129.74, 129.19, 129.02, 128.73, 128.66, 127.96, 127.72, 127.69, 126.71, 126.53, 125.97, 125.29, 123.59, 115.45, 81.52, 68.74, 12.22. HRMS (ESI/Q-ToF): calcd. for C_27_H_20_O_4_ 407.1283, found 407.1284 [M-H]^−^.

### 2.2. Surface Plasmon Resonance

SPR experiments were performed at 25 °C using a Pioneer AE optical biosensor (Sartorius, Varedo, Italy) equipped with a PCH sensor chip (linear polycarboxylate hydrogel layer) and equilibrated with running buffer 10 mM Hepes, pH 7.4, 150 mM NaCl, 0.005% Tween-20, 1% DMSO. The PCH sensor chip was installed and conditioned in accordance with the manufacturer’s instructions. PPARγ-LBD was immobilized on the surface using amine coupling chemistry [39]. Briefly, flow cells were activated for 5 min by injecting 125 μL of 1:1 ratio of 100 mM *N*-hydroxysuccinimide (NHS)/400 mM *N*-ethyl-*N*’-(3-(dimethylamino)propyl)carbodiimide (EDC). The protein solution (150 μg mL^−1^ in 10 mM Na Acetate, pH 4.5) was then injected at 10 μL min^−1^, followed by a 70 μL injection of 1 M ethanolamine (pH 8.0) to block any remaining activated groups on the surface. Approximately 11,770 and 10,800 RUs of protein were immobilized on Channel 1 and Channel 3 of the sensor chip, respectively, whereas Channel 2 was used as a reference for non-specific binding. The stability of the surface was demonstrated by the flat baseline achieved at the beginning (0–60 s) of each sensorgram. The analytes were dialyzed in the running buffer (1% final DMSO concentration) and primary screened using OneStep^®^ injection technology, which allows the compounds to diffuse into a moving stream of buffer to generate a concentration gradient (from 0 up to 10 μM) during a single injection at a flow rate of 50 μL min^−1^ [40]. For each analyte injection, a long dissociation (600 s) was allowed, and regeneration of the sensing surface was not used. Next, the full kinetic analysis of the selected compounds was performed at different serial concentrations (from 5 μM to 312 nM) for 120 s at a constant flow rate of 50 μL min^−1^. For each analyte injection, a dissociation of 180 s was allowed, followed by a short mild regeneration step with 1 M NaCl. All sensorgrams were processed using double referencing. First, responses from the reference surface (Channel 2) were subtracted from the binding responses collected over the reaction surfaces to correct for bulk refractive index changes between the flow buffer and analyte sample. Second, the response from the closest blank injection (zero analyte concentration) was subtracted to compensate for drift and small differences between the activated channel and the reference flow cell Channel 2. As an internal control for the gradient dispersion process, an injection of 3% sucrose dissolved in the assay buffer was performed for all the injections to ensure proper gradient formation and to calibrate for buffer viscosity. To obtain kinetic rate constants and affinity constants, the corrected response data were fitted in QDAT software provided with the instrument (BioLogic Software, Canberra, Australia). A kinetic analysis of the ligand/analyte interaction was obtained by fitting the response data to a reversible 1:1 bimolecular interaction model. The equilibrium dissociation constant (K_D_) was determined by the ratio K_off_/K_on_. This experiment was performed in duplicate.

### 2.3. Transactivation Assay

Reference compounds, the cell culture medium, and the reagents were purchased from Sigma-Aldrich (Merck KGaA). The expression vectors bearing the chimeric receptor containing the yeast Gal4-DNA binding domain fused to the human PPARγ-LBD, and the reporter plasmid for this Gal4 chimeric receptor (pGal5TKpGL3), comprising five repeats of the Gal4 response elements upstream of a minimal thymidine kinase promoter adjacent to the coding sequence for luciferase, were described in a previous study [41]. A culture of the human hepatoblastoma cell line HepG2 (Interlab Cell Line Collection, Genoa, Italy) was conducted in minimum essential medium (MEM) containing 10% heat-inactivated fetal bovine serum, 100 U of penicillin G mL^−1^, and 100 μg mL^−1^ of streptomycin sulfate at 37 °C in a humidified atmosphere of 5% CO_2_. For transactivation assays, cells were seeded in a 24-well plate at a concentration of 10^5^ cells per well and were transfected after 24 h by CAPHOS, a calcium phosphate method, according to the manufacturer’s guidelines. Cell transfection was performed using expression plasmids encoding the fusion protein Gal4-PPARγ-LBD (30 ng), pGal5TKpGL3 (100 ng), and pCMVβgal (250 ng). Following transfection, cells were incubated for 4 h, after which they underwent treatment with the indicated ligands in triplicate for 20 h. Cell extracts were subsequently analyzed for luciferase activity via luminometry (VICTOR^3^ V multilabel plate reader; PerkinElmer, Waltham, MA, USA). *ortho*-Nitrophenyl-β-D-galactopyranoside was used to measure β-galactosidase activity, following a previously described method [42]. All transfection experiments were performed at least twice.

### 2.4. Protein Expression and Purification

PPARγ-LBD (NP_001120802; aa 207−474) was expressed as an N-terminal His-tagged protein using a pET-28a(+) vector and then purified as previously described [43]. Briefly, freshly transformed *E. coli* BL21(DE3) were grown in LB medium with 50 μg mL^−1^ of kanamycin at 37 °C to an OD of 0.7. The culture was then induced with 0.2 mM isopropyl-β-D-thio-galactopyranoside (IPTG) and further incubated at 18 °C for 20 h. Cells were harvested and resuspended in Buffer A (20 mM Tris, pH 8.0, 150 mM NaCl, 1 mM TCEP, 10 mM imidazole, 10% *v/v* glycerol) in the presence of protease inhibitors (Complete Tablets EDTA-free; Roche Applied Science, Penzberg, Germany). Cells were sonicated, and the soluble fraction was isolated by centrifugation. The supernatant was loaded onto a nickel NTA agarose bulk resin (ABT) and eluted with a gradient of imidazole 10–300 mM in Buffer A (batch method). The pure protein was identified by SDS-PAGE. The protein was dialyzed over buffer A to remove imidazole; then, the His-tag was cleaved with thrombin protease (GE Healthcare, Chicago, IL, USA) (10 units/mg) at room temperature for 2 h. The digested mixture was reloaded onto a Ni^2+^-nitriloacetic acid column to remove the His-tag and the undigested protein. The flow-through was dialyzed with buffer B (20 mM Tris, pH 8.0, 1 mM TCEP, 10% *v/v* glycerol, 1 mM TCEP) and then loaded onto a Q-Sepharose HP column (GE Healthcare) and eluted with a gradient of NaCl 0–500 mM in Buffer B using a BioLogic DuoFlow FPLC system (Bio-Rad Laboratories, Hercules, CA, USA). Finally, PPARγ-LBD was purified by gel filtration on a Superdex 75 column (GE Healthcare) and eluted with Buffer C (20 mM Tris, pH 8.0, 1 mM TCEP, 0.5 mM EDTA). For crystallization, the protein was concentrated at 8 mg mL^–1^ using Amicon centrifugal concentrators with a 10 kDa cutoff membrane (Millipore, Burlington, MA, USA).

### 2.5. Crystallization, Data Collection, and Structure Determination

Crystals of apo-PPARγ-LBD were obtained by vapor diffusion at 20 °C using a sitting drop formed by mixing 2 μL of protein solution with 2 μL of reservoir solution (0.8 M Na citrate, 0.15 M Tris, pH 8.0). The crystals were soaked for 3 days in storage solutions (1.2 M Na citrate, 0.15 M Tris, pH 8.0) containing the ligands at a final concentration of 0.5 mM. The ligand dissolved in DMSO (50 mM) was diluted in the storage solution so that the final concentration of DMSO was 2.5%. The storage solution with glycerol 20% (*v/v*) was used as a cryoprotectant. PPARγ-LBD crystallized in the space group C2 (average crystal size: 0.20 mm × 0.20 mm); cell parameters are shown in Appendix A.

X-ray data of PPARγ-LBD in complex with ligands **1** and **3** were collected at 100 K under a nitrogen stream using synchrotron radiation (beamline ID30A-3 at ESRF, Grenoble, France). For the PPARγ-LBD/**1** complex, the diffracted intensities were processed using the programs XDS [44] and Aimless [45], whereas data for the PPARγ-LBD/**3** complex were processed and scaled using the program Xia2 [46]. For both complexes, structure solution was performed with AMoRe [47], using the coordinates of PPARγ/LT175 [48] as the starting model. The coordinates were then refined with CNS [49,50] and PHENIX [10,51] including data between 57.15 and 2.13 Å for the PPARγ-LBD/**1** complex and between 40.81 and 2.20 Å for the PPARγ-LBD/**3** complex. The statistics of crystallographic data and refinement are summarized in Appendix A. The structures of PPARγ-LBD/**1** and PPARγ-LBD/**3** complexes were deposited in the Protein Data Bank with IDs 8ADF and 8C0C, respectively.

### 2.6. In Vitro Kinase Assay

The assay was performed on PPARγ-LBD, in the apo-form and in the complex with ligands **1**–**3**. For the kinase assay, stock solutions of ligands were prepared by diluting with 100% DMSO to a concentration of 500 μM. The stock solutions were further diluted with 50 mM Tris-HCl, pH 7.5, up to the final concentration of 10 μM and pre-equilibrated overnight at 4 °C with the protein.

The kinase assay was carried out for 30 min at 30 °C in 50 μL of the assay buffer containing 50 mM Tris-HCl, pH 7.5, 800 ng of PPARγ-LBD, 10 μM ligand, 1% DMSO, 2.5 mM MgCl_2_, 50 μM DTT, 500 μM ATP, 40 ng of CDK5/p35 (Sigma-Aldrich code no. SRP5011; Merck KGaA). The experiment was performed in triplicate.

### 2.7. Pro-Q^™^ Diamond Phosphoprotein Gel Stain

Pro-Q^™^ Diamond phosphoprotein gel stain (ThermoFisher, Waltham, MA, USA) provides a convenient method for selectively staining phosphoproteins in acrylamide gels, without the need for blotting or the use of phosphoprotein-specific antibodies. After performing electrophoresis, the gel was stained with Pro-Q^™^ Diamond and analyzed using a Chemidoc Imaging System (Bio-Rad). A further step of gel staining with SYPRO^™^ Ruby total-protein gel stain (ThermoFisher) was performed. Determining the ratio of the signal intensities of Pro-Q^™^ Diamond dye to SYPRO^™^ Ruby dye for each band provides a measure of the phosphorylation level normalized to the total amount of protein. More detailed information on the protocol used in this experiment can be obtained on the ThermoFisher website.

### 2.8. Statistical Analysis

Statistical significance was estimated with one-way ANOVA followed by the Dunnett post hoc test, using GraphPad Prism version 9.0 (GraphPad Software, San Diego, CA, USA). Differences with p-values of less than 0.05 were considered statistically significant.

### 2.9. CDK5 Inhibition Test

The degree of CDK5 inhibition in the presence of compounds **1**, **2**, and **3** was evaluated with the same protocol described above for the in vitro kinase assay. In detail, the test was carried out for 30 min at 30 °C in 50 μL of the assay buffer containing 50 mM Tris-HCl, pH 7.5, 800 ng H2A/H2B, 5 μM ligand (for the calculation of the ligand concentration, the excess amount of compounds **1**, **2**, and **3** possibly not bound to PPARγ was overestimated), 1% DMSO, 2.5 mM MgCl_2_, 50 μM DTT, 500 μM ATP, 40 ng of CDK5/p35 (Sigma-Aldrich code no. SRP5011; Merck KGaA). The Pro-Q^™^ Diamond Phosphoprotein Gel Stain method and a Chemidoc Imaging System (Bio-Rad) were used for measuring the phosphorylation level. The experiment was performed in triplicate and the statistical significance was estimated as above.

## 3. Results and Discussion

We approached this study by investigating the bioactivity of an in-house library of γ-hydroxy-lactone derivatives. We decided to perform a biological screening, through SPR technology-based experiments, with the aim of identifying new scaffolds for the development of innovative anti-diabetic drugs.

The OneStep^®^ injection technology was used for the screening of our compound collection. This method has been successfully employed in our previous work to test a series of natural products, extracted from medicinal plants [52]. Its efficiency and reliability as a screening technique were confirmed by the rapid identification of promising compounds. This approach is based on the well-established concept of Taylor dispersion injection [53], which allows the compound to diffuse into a moving stream of buffer, producing a gradient throughout the injections. In this way, a wide range of analyte concentrations can be readily tested. In this assay, all the ligands from the DMSO stock solutions were dialyzed into the running buffer at 10 μM (1% final DMSO concentration) and gradient-injected for 120 s over the sensor chip surface. With the aim of preserving the functionality of the immobilized receptors throughout the duration of the experiment, a long dissociation time (600 s) was allowed, and no regeneration of the sensing surface was used. Based on the sensorgrams profile and the affinity characterization, three out of the tested compounds (**1**–**3**) were selected for the full kinetic analysis, whereas the compounds showing a bad profile, no binding or low-affinity interactions (K_D_ > 10 μM) were no further characterized. Compounds **1** and **3** were chosen for the good-quality data of both the association and dissociation phases of the sensorgrams, as well as for the low-micromolar affinity range of the binding. By contrast, compound **2**, despite showing a less regular profile, was selected by considering the slower dissociation phase of the interaction, which resulted in a low-micromolar affinity constant. The sensorgrams of a duplicate experiment and the average of the measured affinity constants for all of the screened compounds are represented in Appendix A. The ligand LT175 was used as the reference compound, as its binding affinity had already been measured in our previous works using different techniques, such as SPR (K_D_ = 2.34 μM) and ITC (K_D_ = 3.66 μM) [52,53,54], and confirmed by the OneStep^®^ screening assay (K_D_ = 2.55 μM). After the preliminary screening performed on the whole library, SPR was further employed to measure both the binding affinity (K_D_) and rate constants (k_on_ and k_off_) of **1**–**3** for PPARγ; these values were compared to those of the reference ligand LT175 [48]. PPARγ was immobilized onto the sensor chip, and **1**–**3** were tested at the conditions described in the Experimental Section. All three compounds showed promising K_D_ values, ranging between 1.67 and 3.75 μM, as shown in Figure 3.

Furthermore, compounds **1**–**3** were evaluated in vitro for their agonist activity towards the human PPARγ subtype (hPPARγ) by employing the GAL4-PPAR transactivation assay. The results were compared to the corresponding data for rosiglitazone, used as a reference in the test. The maximum fold induction obtained with the reference agonist was defined as 100%. We evaluated the efficacy of **1**–**3** on PPARγ at increasing concentrations up to 100 μM. As shown in Figure 4B, none of the tested compounds showed a significant agonist activity in the PPARγ transactivation assay, up to the maximum tested dose (100 µM). The compounds were also analyzed in antagonist mode by conducting a competitive binding assay, in which PPARγ activity was measured at a fixed concentration of the reference agonist rosiglitazone (1 µM) in cells treated with increasing concentrations of **1**–**3**. The inhibition curves, from which the IC_50_ values were extrapolated as mean ± SEM of three duplicate experiments, are reported in Figure 4A. All compounds exhibited IC_50_ values in the high-micromolar range, without significant differences among them (Figure 4C). Therefore, considering that **1**, **2,** and **3** bind to PPARγ but do not affect its transcriptional activity, it is reasonable to assert that they can be classified as PPARγ non-agonists. However, these compounds (especially **3**) showed a certain cytotoxicity on the cell line used in the assay (HepG2), at the highest tested concentrations (data not shown; visually examined under a visible light microscope, the cells appeared fewer in number, more rounded and partially detached at concentrations of 25 and 100 µM, with a dose-dependent effect). Consequently, the IC_50_ values could not be estimated with a certainty above 10 µM, which was the threshold at which the cytotoxicity was null or negligible. From the inhibition curves (Figure 4A), we hypothesized the absence of antagonism: the activity remained unchanged up to 10 µM and then decreased at 25 µM and 100 µM. However, at these concentrations, the signal reduction could not be unequivocally attributed to PPARγ antagonism, due to the cytotoxic effects of the compounds at higher doses.

Finally, to assess the ability of **1**–**3** to inhibit the CDK5-mediated phosphorylation of PPARγ in vitro, a kinase assay was set up and optimized to measure the inhibition rate, as described in the experimental section. The phosphoproteins were detected and quantified by using the Pro-Q^™^ Diamond phosphoprotein gel stain method. Compounds **1**, **2**, and **3** effectively inhibited Ser245 phosphorylation by nearly 40%, whereas, for the positive control (the ligand NV1380 [55]), an inhibition slightly higher than 40% was calculated (Figure 4D).

Furthermore, these molecules were also screened as potential inhibitors of CDK5 itself, considering their structural similarity to BLI (Figure 2), which was reported in the literature as both a PPARγ partial agonist and a CDK5 inhibitor [10]. A kinase activity test was performed in the presence of compounds **1**, **2**, and **3**, using histone H2a/H2b as a phosphorylation target. The analysis revealed that, among the three selected molecules, **1** was the only one that showed a very low inhibition effect on the kinase. Conversely, compounds **2** and **3** exerted a more significant inhibitory activity (Figure 4E). Nevertheless, all three compounds showed good interaction characteristics with PPARγ.

Based on these results, we decided to deeply investigate the interaction mode of **1**–**3** with PPARγ-LBD by performing crystallographic experiments on the corresponding ligand/target complexes.

To this end, diffraction data were collected from apo-crystals soaked for 3 days in the presence of the ligand at 0.5 mM. The crystal structures of PPARγ-LBD in complex with **1** and **3** were determined at 2.13 Å and 2.20 Å, respectively, each comprising two PPARγ-LBD monomers in the asymmetric unit. Despite the compounds being synthesized as racemic mixtures, only the (*R*)-enantiomer was observed in the crystals.

The two structures were nearly identical to each other, with an RMSD value of 0.539 Å calculated for Cα atoms, and showed great similarity to that of PPARγ-LBD in complex with rosiglitazone (PDB ID: 4EMA), with an RMSD of 0.519 and 0.807 for **1** and **3**, respectively. Unfortunately, we were not able to solve the structure of the PPARγ–LBD/**2** complex because the electron density map was not clear enough to allow the unambiguous positioning of the ligand.

The omit map of compound **1**, calculated from the refined model, clearly showed the electron density around the ligand (Figure 5A), occupying the canonical partial agonist hydrophobic binding region between helix 3 (H3) and the *β*-sheets of PPARγ-LBD [56]. Surprisingly, compound **1** showed two distinct binding modes within the two monomers; however, its position inside molecule B seems to be the most effective for the inhibition of Ser245 phosphorylation. Indeed, no interactions between compound **1** and any residues belonging to the *β*-sheet of molecule A were observed (Figure 5D). On the contrary, within molecule B, the oxygen atom of the lactone group formed a hydrogen bond with the Oγ atom of Ser342 (H-bond distance 3.1 Å), and the oxygen atom of the hydroxyl group established a further H-bond with the backbone NH of the same serine residue (H-bond distance 3.5 Å). Both the oxygen atoms from the lactone group and the nitrogen atom of Ser342 were also involved in an additional H-bond with the water molecule 205. Inside molecule B of the asymmetric unit, the hydroxyl group of the phenol moiety formed a hydrogen bond with the oxygen atom of Arg280 (H-bond distance 3.4 Å), while the bromine atom on the opposite aromatic ring was involved in a halogen bond with the sulfur atom of Met364 (distance 3.4 Å). The ligand also interacted with Cys285, Ile281, Met348, Leu 255, and Gly284 through hydrophobic contacts (Figure 5A).

Conversely, while occupying the same hydrophobic region of **1**, compound **3** was observed only in molecule B of PPARγ-LBD. As the ligand was modeled with an occupancy of 0.78, the electron density arising from the omit map was relatively weak and it was probably masked by the bulk-solvent model. For this reason, the interpretative power of the residual map was improved by preventing bulk solvent from entering the selected omit region using the Polder (omit) map approach available in the Phenix software suite [57]. The statistical analysis of the Polder (omit) map showed that the calculated F_obs_ without the ligand (CC_1,3_ = 0.7153) was larger than the calculated F_obs_ with the ligand (CC_1,2_ = 0.6629), thus supporting the presence of the ligand in the *β*-sheet sub-pocket of the LBD. In this way, the weak density became visible, and the ligand was clearly modeled inside the electron density (Figure 5C). As described by the superposition between the two ligands, compound **3** was more deeply positioned within the hydrophobic binding pocket (Figure 5E); moreover, the side chain of Arg288 was shifted in the PPARγ–LBD/**3** complex and involved in a cation-π interaction with the naphthalene ring of the ligand. The ligand interacted with Ser342 through a hydrogen bond between the carbonyl oxygen of the lactone moiety and the NH of the serine backbone. A further hydrogen bond was formed between the hydroxyl group and the water molecule 120. The positioning of the ligand inside the pocket was also facilitated by many hydrophobic interactions involving residues Phe226, Leu228, Cys285, Ala292, Glu295, Met329, Leu330, Leu333, Leu340, Ile341, and Met364 (Figure 5B).

As evidenced by the two-dimensional scheme of the binding mode depicted in Figure 6 and calculated using Ligplot+ [58], both **1** and **3** were involved in a direct hydrogen bond with Ser342, thus contributing to the stabilization of the β-sheet. Interestingly, both the stabilization of Ser342 and the ligand binding in the hydrophobic pocket of PPARγ-LBD have been previously described as possible inhibitory mechanisms of the CDK5-mediated PPARγ phosphorylation at Ser245 [59,60]. Furthermore, the non-agonist profile of these compounds could be ascribed to their positioning between H3 and the *β*-sheet, far from H12 whose stabilization is essential for the recruitment of the coactivators and the triggering of the transcription machinery. On the other hand, these compounds, while contracting only a few interactions within the LBD, interacted with Ser342, which is located right on the *β*-sheet, close to Ser245, whose stabilization probably explains the inhibitory effect on the phosphorylation. An analogous situation has also been observed with other compounds that are non-agonists or weak agonists of PPARγ (e.g., MRL24, PDB ID: 2Q5P; SR1664, PDB ID: 4R2U; BLI, PDB ID: 6L89).

## 4. Conclusions

In this work, we identified the γ-hydroxy-lactone derivatives **1**–**3** as new inhibitors of CDK5-mediated PPARγ phosphorylation. The investigation of the interaction between the ligands and PPARγ-LBD, performed at a molecular level by crystallographic experiments, will provide better opportunities to design novel PPARγ partial agonists or non-agonists with potential therapeutic applications. Until recently, it was a belief that the pharmacological action of PPARγ ligands resulted from their functional agonism on this receptor. Now, it is clear that classical agonism is not required for PPARγ ligands to exert a strong anti-diabetic action. In fact, a non-agonist profile may be even desirable to reduce or eliminate the typical side effects related to PPARγ activation, which have so far limited its use as a therapeutic target. In this study, we proved that the γ-hydroxy-lactone scaffold may provide a promising template for the development of novel anti-diabetic agents targeting PPARγ, without any transactivation potential or agonist activity. We observed that these compounds show a certain degree of cytotoxicity, but this occurs only at concentrations considerably higher than the measured affinity constant K_D_.

Our data suggest that compound **1** prevents Ser245 phosphorylation by acting mainly on PPARγ stabilization and exerting a very weak CDK5 inhibitory effect. Conversely, the activity of compounds **2** and **3** could be ascribed mainly to a direct CDK5 inhibition rather than PPARγ stabilization, although all three compounds bind PPARγ with good affinity. For this reason, among the tested molecules, **1** emerged as the most interesting one. Differently from BLI, this compound showed a minimal effect on the phosphorylation of a CDK5 substrate (H2a/H2b), which means that it should not disrupt the basal function of CDK5. CDK5 holds vital biological and clinical significance because it regulates various cellular signaling pathways. The CDK5-p35 complex is involved in the phosphorylation of different substrates leading to its activation in normal conditions; therefore, interfering with its action may have serious consequences determined by systemic side effects.

Hence, considering its synergistic therapeutic potential and synthetic feasibility, **1** could represent a promising starting point for the development of new derivatives as innovative drugs for the treatment of metabolic disorders.

## Data Availability

The raw data supporting the conclusions of this article will be made available by the authors, without undue reservation.

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
