# Peer review of "Biological Screening and Crystallographic Studies of Hydroxy γ-Lactone Derivatives to Investigate PPARγ Phosphorylation Inhibition"

_biomolecules, 2023, doi:10.3390/biom13040694_

Round 1

Reviewer 1 Report

The manuscript “Biological screening and crystallographic studies of hydroxy γ-lactone derivatives to investigate PPARγ phosphorylation inhi-bition”,

The paper highlights on the concept of non-agonist activity to PPARγ, as a therapeutic target for the treatment of type II diabetes. In this context, the authors synthesised and tested a new family of compounds with the hydroxy γ-lactone heterocycle system. The article presents a multidisciplinary approach and a wide variety of techniques correctly employed. In order to improve and clarify the work, I propose the following:

1. The authors describe sufficiently the biochemical pathway of PPAR/CDK5 complex phosphorylation and its relationship with insulin sensitivity, however it would be useful to include a simple figure diagram to facilitate the understanding of this process.

2. Since PPARγ is an emerging target for the treatment of metabolic syndrome, I consider that it would be interesting for the authors to reference research reporting agonists, such as Li, Z., Ren, Q., Zhou, Z., Cai, Z., Wang, B., Han, J., & Zhang, L. (2021). Discovery of the first-in-class dual PPARδ/γ partial agonist for the treatment of metabolic syndrome. European Journal of Medicinal Chemistry, 225, 113807.

3. When the authors said "ligands to PPARγ was evalu-ated by Surface Plasmon Resonance (see figure S2 SI)" it should be changed to Table S2.

4. The quality of the sensograms in figure 2 is very low, I suggest to improve it or leave them in the supplementary section.

5. Compounds 7 and 9 can be present as tautomers in the keto or enol form, I suggest to keep the same criteria for all families and show only the keto form, unless the authors have a reason to the contrary.

6. A number of synthetic intermediates have been previously reported, the authors should reference them.

7. The text in the section of characterisation and synthesis of compounds should be revised and improved

8. The authors note that they started from a racemic mixture for the crystallisation of the ligand-enzyme complex, then how is it possible that the crystallographic studies show only one enantiomer?; the authors must argue sufficiently.

9. The authors should present the cytotoxicity study results, especially because the authors indicate that they showed partial toxicity.

10. Authors should review the style of the references and to avoid the use of et al.

Author Response

Reviewer #1: The manuscript “Biological screening and crystallographic studies of hydroxy γ-lactone derivatives to investigate PPARγ phosphorylation inhi-bition”,The paper highlights on the concept of non-agonist activity to PPARγ, as a therapeutic target for the treatment of type II diabetes. In this context, the authors synthesised and tested a new family of compounds with the hydroxy γ-lactone heterocycle system. The article presents a multidisciplinary approach and a wide variety of techniques correctly employed. In order to improve and clarify the work, I propose the following:

  1. The authors describe sufficiently the biochemical pathway of PPAR/CDK5 complex phosphorylation and its relationship with insulin sensitivity, however it would be useful to include a simple figure diagram to facilitate the understanding of this process.

We thank the Reviewer for the suggestion. We added a new figure in the introduction.

  1. Since PPARγ is an emerging target for the treatment of metabolic syndrome, I consider that it would be interesting for the authors to reference research reporting agonists, such as Li, Z., Ren, Q., Zhou, Z., Cai, Z., Wang, B., Han, J., & Zhang, L. (2021). Discovery of the first-in-class dual PPARδ/γ partial agonist for the treatment of metabolic syndrome. European Journal of Medicinal Chemistry, 225, 113807.

We added the references regarding PPARα/γ and PPAR δ/γ dual agonists.

  1. When the authors said "ligands to PPARγ was evalu-ated by Surface Plasmon Resonance (see figure S2 SI)" it should be changed to Table S2.

We changed to Table S2.

  1. The quality of the sensograms in figure 2 is very low, I suggest to improve it or leave them in the supplementary section.

We improved the quality of Figure 2.

  1. Compounds 7 and 9 can be present as tautomers in the keto or enol form, I suggest to keep the same criteria for all families and show only the keto form, unless the authors have a reason to the contrary.

We thank the Reviewer for the suggestion. We indicated both compounds 7 and 9 in the enol form, considering that the NMR spectrum of 9 shows that the compound is actually an enol.

  1. A number of synthetic intermediates have been previously reported, the authors should reference them.

Following the suggestion of the Reviewer, we added the most relevant references to the chemical procedures.

  1. The text in the section of characterisation and synthesis of compounds should be revised and improved

The section has been revised.

  1. The authors note that they started from a racemic mixture for the crystallisation of the ligand-enzyme complex, then how is it possible that the crystallographic studies show only one enantiomer?; the authors must argue sufficiently.

Actually, on the basis of our experience, it is not surprising that only one enantiomer of the racemic mixture is observed in the crystallographic structure of the complex. Indeed, the electron density map of the first refinement was quite clear and it was not possible to fit the S-enantiomer.

  1. The authors should present the cytotoxicity study results, especially because the authors indicate that they showed partial toxicity.

We have highlighted in the text the part related to the cytotoxic effects observed for the compounds. This is a qualitative analysis, which, nonetheless, has a significant value. An actual cytotoxicity experiment would definitely take much longer than the time we have been given to respond.

  1. Authors should review the style of the references and to avoid the use of et al.

We thank the Reviewer for the suggestion; however, the MDPI Citations Style Guide states that for documents co-authored by a large number of persons (more than 10 authors), the first ten authors can be indicated, followed by ‘et al.’ at the end.

Reviewer 2 Report

Dr. Davide Capelli et al shown a novel kind of Hydroxy gamma-lactone derivatives in inhibition of PPARr phosphorylation using biological screening and crystallographic studies. This manuscript provides a great pattern for drug design, especially for safer anti-diabetic drug. A few questions are as following:

1. I may suggest to include a more comprehensive introduction about the PTMs of PPARgamma (not only phosphorylation, such as acetylation,SUMOylation,...)  especially for the potential of safer drug design.

2. Are there any greater potential for Compound 1,2,3 compared to NV1380 or BLI?

3. In the Figure 3D, it is better to show the WB of total PPARgamma as well as PPARgamma Ser245. 

Author Response

RESPONSES TO REFEREES’ COMMENTS: Referees’ comments are in black, our responses in blue.

Reviewer #2: Dr. Davide Capelli et al shown a novel kind of Hydroxy gamma-lactone derivatives in inhibition of PPARr phosphorylation using biological screening and crystallographic studies. This manuscript provides a great pattern for drug design, especially for safer anti-diabetic drug. A few questions are as following:

Thanks to reviewer #2 for the suggestions.

  1. I may suggest to include a more comprehensive introduction about the PTMs of PPARgamma (not only phosphorylation, such as acetylation,SUMOylation,...) especially for the potential of safer drug design.

We implemented the introduction with a general description of the importance of all PTMs in PPARγ activities and briefly described how PTMs could be exploited for the drug discovery process.

  1. Are there any greater potential for Compound 1,2,3 compared to NV1380 or BLI?

In the manuscript, NV1380 was used as a reference compound in the kinase assay because of its strong inhibitory effect on PPARγ phosphorylation by CDK5, as demonstrated in our previous work.

According to the literature, BLI is also an inhibitor of PPARγ phosphorylation, even if its inhibitory activity could be mostly ascribed to the direct interaction with CDK5 rather than with the receptor. On the contrary  compounds 1-3, and especially 1, while sharing with BLI the same lactone moiety, showed a lower inhibition effect on the kinase and, for this reason, they can be considered in this stage as promising scaffolds to be implemented in further studies of rational drug design.

  1. In the Figure 3D, it is better to show the WB of total PPARgamma as well as PPARgamma Ser245. 

All the images of the gels have been sent to the Editor during the submission process. Anyway, as described in the experimental section, the experiment is not a WB but a different assay in which a specific phopsho-protein gel stain has been used. This experiment is based on two steps of staining:

  1. Staining of phospho-proteins;
  2. Staining of total protein.

The second step is aimed to normalize the bands to the total amount of protein, in order to compensate for the manual error in the gel sample loading. 

Round 2

Reviewer 1 Report

I consider that the paper was improved and satisfies the journal's standards to be published.